# Contact tracing reveals community transmission of COVID-19 in New York City

Sen Pei [1] ✉, Sasikiran Kandula[1], Jaime Cascante Vega [1], Wan Yang[2], Steffen Foerster[3], Corinne Thompson[3], Jennifer Baumgartner[3], Shama Desai Ahuja[2,3], Kathleen Blaney [3], Jay K. Varma [4], Theodore Long[5] & Jeffrey Shaman [1,6]

Understanding SARS-CoV-2 transmission within and among communities is critical for tailoring public health policies to local context. However, analysis of community transmission is challenging due to a lack of high-resolution surveillance and testing data. Here, using contact tracing records for 644,029 cases and their contacts in New York City during the second pandemic wave, we provide a detailed characterization of the operational performance of contact tracing and reconstruct exposure and transmission networks at individual and ZIP code scales. We find considerable heterogeneity in reported close contacts and secondary infections and evidence of extensive transmission across ZIP code areas. Our analysis reveals the spatial pattern of SARS-CoV-2 spread and communities that are tightly interconnected by exposure and transmission. We find that locations with higher vaccination coverage and lower numbers of visitors to points-of-interest had reduced within- and cross-ZIP code transmission events, highlighting potential measures for curtailing SARS-CoV-2 spread in urban settings.

Within metropolitan areas, infection risk and disease burden due to SARS-CoV-2, the causative agent of COVID-19, are characterized by spatial heterogeneity at neighborhood scales[1–3]. Communities with substantial local infections can sustain the spread of SARS-CoV-2, seed infections in interconnected neighborhoods, and spark resurgences of cases following the relaxation of non-pharmaceutical interventions (NPIs), such as masking and social distancing[4]. In densely populated urban settings, public health tactics may need to be uniquely tailored to specific geographic areas and/or communities that most support the persistence and spatial dispersion of SARS-CoV-2 infections. Development of such tailored tactics requires improved understanding of both transmission patterns at fine geographical scales and the factors shaping the intensity of community outbreaks. Examples of previously utilized targeted interventions include limiting indoor dining and gathering, increasing testing availability, encouraging

home quarantine for exposed contacts, requiring face masks indoors, and closing nonessential businesses in high-risk communities. While the transmission patterns of SARS-CoV-2 at global, national, and regional levels have been reported[5–13], research on community-level transmission is often challenging due to limited availability of high-resolution surveillance and testing data, the lack of routine case interviews, and the difficulty identifying transmission events. In addition, the effect of public health interventions on community transmission of SARS-CoV-2 in metropolitan areas has not been well evaluated.

Data collected through contact tracing efforts have provided valuable insights into the transmission dynamics of SARS-CoV-2;[14–18] however, most contact tracing during the early phase of the pandemic mainly focused on specific local outbreaks, which cannot support population-level analysis of community transmission. Here, we use

---

[1]Department of Environmental Health Sciences, Mailman School of Public Health, Columbia University, New York, NY 10032, USA. [2]Department of Epidemiology, Mailman School of Public Health, Columbia University, New York, NY 10032, USA. [3]New York City Department of Health and Mental Hygiene (DOHMH), Long Island City, NY 11001, USA. [4]Department of Population Health Sciences, Weill Cornell Medical College, New York, NY 10065, USA. [5]NYC Health + Hospitals, New York, NY, USA. [6]Columbia Climate School, Columbia University, New York, NY 10025, USA. ✉e-mail: sp3449@cumc.columbia.edu

detailed data from confirmed and probable cases[19] and case investigations during the second pandemic wave in New York City (NYC) to quantify community spread of COVID-19 at small spatial scales from October 2020 to May 2021. Unlike the initial outbreak during the spring of 2020, the second pandemic wave was fully captured by contact tracing. Additionally, contact tracing operation and individual protective measures such as mask-wearing and social distancing remained relatively stable during this period of the pandemic (in contrast with the post-Omicron era when protective measures were largely abandoned). As a result, data collected during the second pandemic wave may better inform understanding of SARS-CoV-2 community transmission in NYC and the operational performance of contact tracing during a public health emergency.

## Results

### Contact tracing in NYC

The NYC Test & Trace Corps initiative was launched in June 2020[20]. Established as an operation to provide contact tracing, testing, and resources to support isolation and quarantine, the contact tracing program was integrated with a set of intervention efforts designed to limit morbidity and mortality from COVID-19 in NYC (Supplementary Information). Contact tracing was performed through phone calls and text messages, capable of reaching most residents of NYC. Specifically, contact tracers made phone calls to confirmed cases and symptomatic contacts to conduct a case investigation. For children under 18 years old, parents or legal guardians were contacted. Information about close contacts during the infectious period was elicited during the interview, and reported close contacts were then notified about their status of exposure through phone calls or text messages and are encouraged to get tested. Both confirmed/probably cases and their close contacts were monitored daily for the duration of their quarantine.

We analyzed data obtained from case investigations and COVID-19 testing results (molecular and antigen) collected between 1 October 2020 and 10 May 2021 (Supplementary Fig. 1, Supplementary Information). During this period, 691,834 confirmed and probable cases were reported to the New York City Department of Health and Mental Hygiene (DOHMH)[21]. The circulating strains of SARS-CoV-2 in NYC were dominated by the index virus strain; however, the Iota (B.1.526) and Alpha (B.1.1.7) variants gradually replaced the index virus during the spring of 2021 (Supplementary Fig. 2). After excluding cases residing in residential congregate settings, cases were sent to the NYC

Test & Trace Corps for contact tracing. Among these cases, 644,029 were reached by tracers and 450,415 completed an interview. In total, 779,011 contacts with confirmed and probable cases were self-reported via case investigations, of whom 20.9% (162,659/779,011) were subsequently tested. The overall positivity rate among tested exposures is 55.8%. However, as infected individuals were more likely to seek tests, the actual secondary attack rate should be lower. We further disaggregated testing results for different exposure types (healthcare facility contact, home health aide, household member, intimate partner, large gathering contact, other close proximity, workplace contact) (Supplementary Fig. 3). The positivity rate was highest for household members and lowest for workplace contacts. The median time from specimen collection to reporting results to DOHMH was 2 days. 97% of index patients were called by tracers within two days of reporting to DOHMH (Fig. 1a) and 68.4% of contacts were called the day of reporting to the Test & Trace team (Fig. 1b). Among tested contacts, 66.6% sought testing within one week of notification (Fig. 1c). For traced symptomatic infections, 86.7% were tested after symptom onset, and 13.3% were tested before symptom development (Fig. 1d).

Adults aged 20 to 49 years old constituted the majority of index cases (Fig. 1e), a finding in agreement with the age distribution of confirmed infections in the United States[22]. Self-reported contacts were more uniformly distributed among the population under 50 years old (Fig. 1f). The age-stratified contact matrix highlights more frequent interactions among individuals of similar age and inter-generation mixing within the household (Fig. 1g), a pattern also observed in other countries[23].

### Exposure and transmission networks

We reconstructed the self-reported exposure network at the individual level for the study period. The exposure network was highly fragmented, with 947,042 individuals in 242,486 disjoint clusters. Cluster size showed considerable heterogeneity (Fig. 2a), as did the number of contacts reported by each index case (Fig. 2b). We visualize several large exposure clusters in Fig. 2c, color-coded by the home borough of each person. Exposure clusters exhibit diverse structures ranging from hub-and-spoke networks with a single spreader to networks with multiple spreaders. Over half of the clusters shown in Fig. 2c were in Queens and Brooklyn. Within those large exposure clusters in Fig. 2c, 1195 index patients (59.4%) reported contacts living in the same borough, but 817 (40.6%) cross-borough contacts were also recorded.

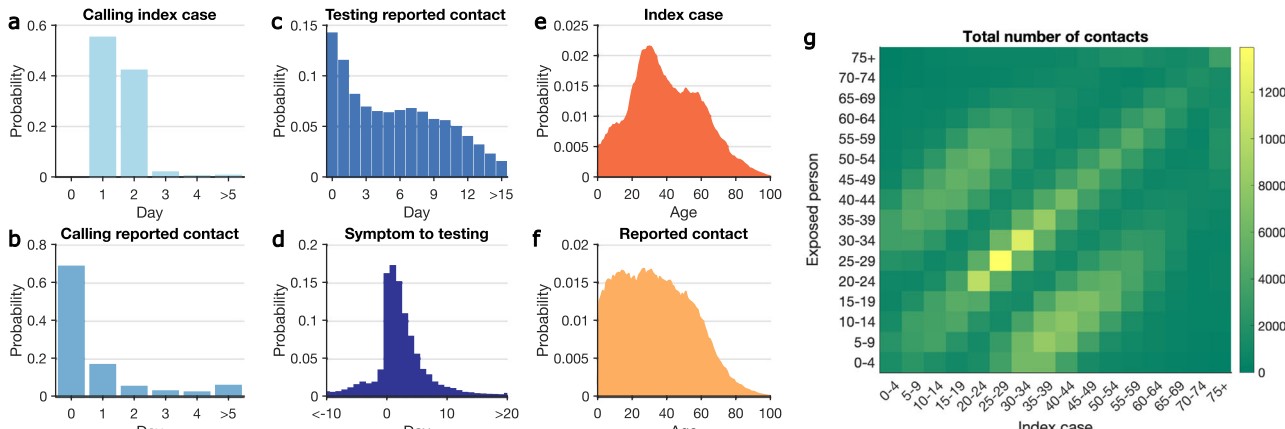

**Fig. 1 | Key statistics of contact tracing in NYC. a–d** The distributions of: **a** time between reporting date for index cases and being called by contact tracers; **b** time between calling index cases and notifying exposed persons; **c** time between notifying exposed persons and specimen sampling of notified individuals who were tested; **d** time from symptom onset to specimen sampling for symptomatic COVID infections. A negative value implies that testing preceded symptom onset. Age distributions of index cases (**e**) and self-reported contacts (**f**). The contact mixing matrix (**g**) shows the total number of exposures among age groups reported during the study period.

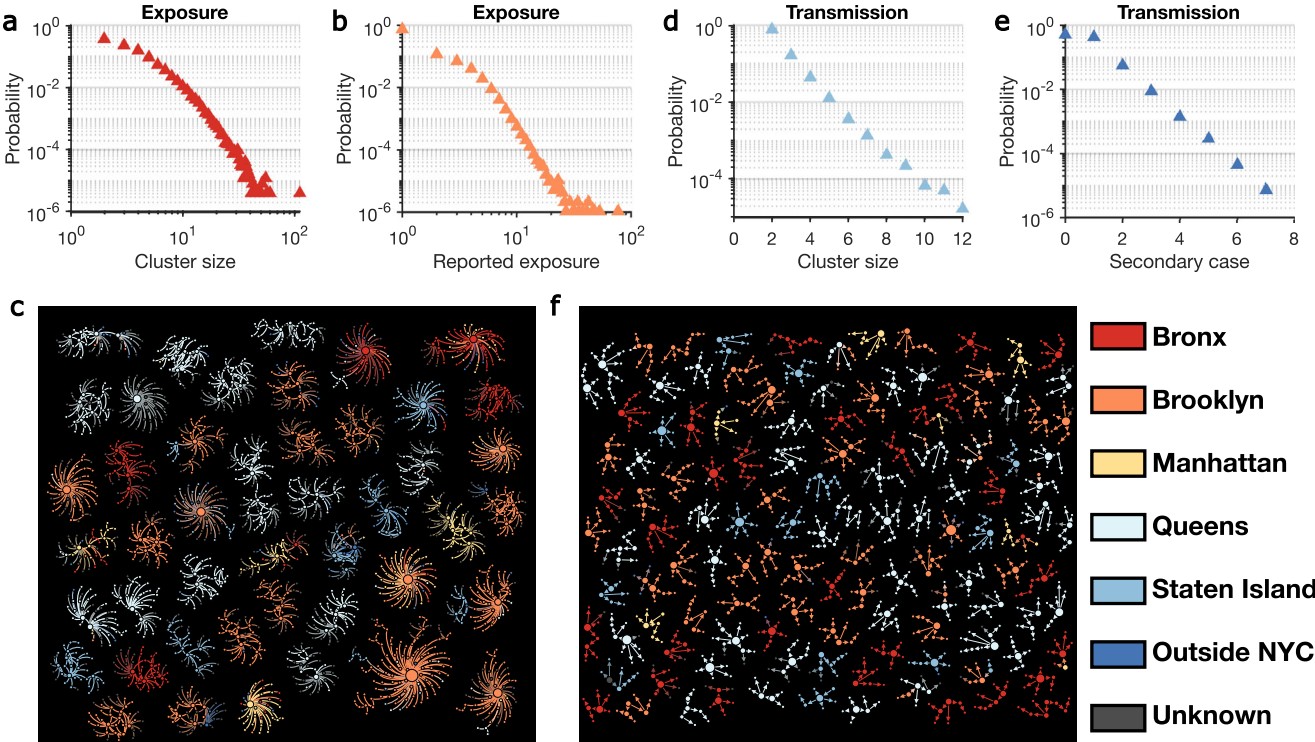

**Fig. 2 | Structure of exposure and transmission networks. a, b** The distributions of cluster size and number of close contacts reported by each index case in the exposure network. Exposure clusters with more than 35 individuals are visualized in (**c**). The exposure network is undirected. Index cases and reported close contacts are connected. Node size is proportional to the number of connected individuals. Colors indicate the home location of each person (five boroughs in NYC, outside NYC, and unknown). The distributions of cluster size and the number of secondary cases in the transmission network are shown in (**d**) and (**e**), respectively. **f** Visualizes transmission clusters with more than six infected individuals. Node size represents the number of secondary cases. Arrows indicate the direction of transmission.

We additionally reconstructed transmission chains between index cases and their close contacts who were confirmed positive in laboratory tests (molecular and antigen). Due to asymptomatic and pre-symptomatic shedding[24–26], index cases were not necessarily the source of infections in these putative transmission events. To infer the direction of transmission, we estimated the infection date of lab-positive cases. For symptomatic cases, infection date was estimated using an empirical incubation period distribution obtained from a prior study[18]; for asymptomatic cases, we used specimen collection date to estimate infection date using a model of viral load dynamics coupled with a Bayesian inference (Supplementary Fig. 4)[27]. Specifically, for each index case and close contact pair, we estimated their infection times using symptom onset date or specimen collection date. The direction of transmission was then determined by the estimated infection times—the individual infected earlier is the infector and the individual infected later is the infectee. We sampled an ensemble of possible transmission networks compatible with the estimated chronological order of infections. For each sampled transmission network, we computed the likelihood of observing the network given transmission probabilities across age groups, estimated using the test and trace data (Supplementary Table 1, Supplementary Fig. 5). The reconstructed network was selected as the one that maximizes the likelihood among the ensemble of possible transmission networks. We further performed sensitivity analyses demonstrating that the network reconstruction is robust to potential bias of the incubation period distribution[28] (Supplementary Information). More details on the transmission network reconstruction are provided in the Supplementary Information.

During the study period, we identified 58,474 potential transmission clusters formed by exposures that resulted in lab-confirmed infections. On average, these transmission clusters had a mean size of 2.3 individuals, representing 19.6% (135,478/691,834) recorded cases during the study period. However, transmission cluster size and the number of secondary cases linked to each index case had large variance (Fig. 2d, e)—only 0.2% of transmission clusters involved more than 6 infections. The largest identified transmission cluster consisted of 12 cases, and the maximum number of secondary cases for a single index case was 7. Transmission clusters with at least 6 infections are visualized in Fig. 2f.

To quantify the spatial spread of SARS-CoV-2 in NYC at fine geographical scales, we mapped exposure and transmission networks across modified ZIP code tabulation areas (MODZCTAs, referred to as ZIP codes hereafter; Fig. 3a, b). Among 72,191 transmission events where place of residence was known, 7826 (10.8%) included multiple ZIP codes. Among these cross-ZIP code transmission events, only 2536 (32.4%) occurred between neighboring ZIP code areas, indicating that the majority of cross-ZIP code transmission drove non-local disease spread. For 2187 cross-borough transmission events, only 48 (2.2%) were between neighboring ZIP code areas. We observed several local clusters of ZIP codes that were tightly interconnected by exposure and transmission, centered around locations with high community prevalence. Infections in those high-prevalence ZIP code clusters were linked to self-reported contacts in nearby and far locations (Fig. 3a), which may have facilitated the spread of COVID-19 across the city (Fig. 3b). Among the cross-ZIP code transmission chains, we examined distributions of index cases who initiated transmission (Fig. 3c) and the infected contacts (Fig. 3d) across ZIP codes. A distinct skew in the distribution suggests that certain ZIP codes were more involved in the spatial spread of COVID-19. Geographically, most cross-ZIP code transmission events occurred within 10 km; however, long-distance transmission up to 40 km was also evident (Fig. 3e).

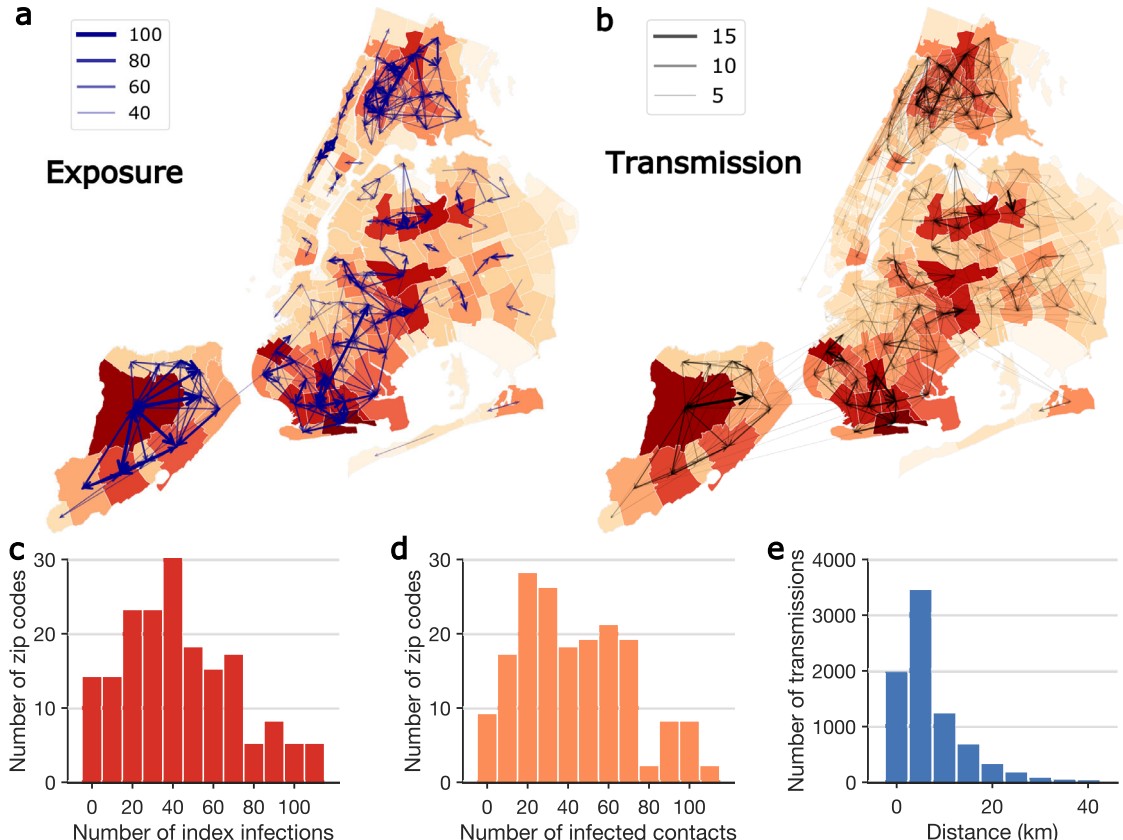

**Fig. 3 | Spatial transmission of SARS-CoV-2 in NYC. a, b** The exposures and transmission events across ZIP codes in NYC identified from contact tracing data. Arrows indicate direction of exposure (from index cases to reported close contacts) and transmission (from index infections to infected contacts). Arrow thickness indicates the number of exposures and transmission events. ZIP code area color represents the cumulative number of confirmed cases during the study period (yellow to red—low to high). To better visualize, exposure links with less than 30 events and transmission links with <2 events are not shown on the maps. For cross-ZIP code transmission events, the distributions of index infections and infected contacts across ZIP code areas are presented in **c** and **d. e** The distribution of distance between home ZIP codes of index infections and infected contacts in cross-ZIP code transmission events. The population weighted centroids for ZIP code areas were used to compute the distance.

## Evaluation of intervention measures

During the period from October 2020 to March 2021, a dynamic zone-based control strategy was adopted in New York State to limit viral spread in communities with high case growth rates while avoiding undue harm to the economy[29]. Three tiers of zones (yellow, orange, and red) were identified based on a set of metrics, collectively defined by test positivity rate, hospital admissions per capita, and hospital capacity[29,30]. Local restrictions on business and services were imposed based on zone conditions. Compliance to these restrictions can be reflected by the number of individuals visiting points-of-interest (POIs, e.g., restaurants, grocery stores, gyms, and bars) in each ZIP code. In December 2020, vaccines became available to the population at highest risk for severe outcomes associated with COVID-19 in NYC and were subsequently available to all eligible individuals over 15 years old during early April 2021. With the support of the detailed contact tracing data, we evaluated the impact of these public health interventions on community transmission of SARS-CoV-2 in NYC.

We assessed the associations of the numbers of non-household within- and cross-ZIP code transmission events across NYC with demographic, socioeconomic, disease surveillance, vaccination coverage, and human mobility features (Supplementary Information, Supplementary Figs. 6–7). Here cross-ZIP code transmission events include both directions, i.e., transmission for which either infector or infectee lived in a certain ZIP code. As non-household transmission contributed to the expansion of SARS-CoV-2 outside the household, we focused on 4642 non-household transmission events, representing 7% of all transmission events. We used aggregated foot traffic records

derived from mobile phone data[31] documenting weekly numbers of POI visitors in each ZIP code as an indicator of human mobility and compliance with the zone-based local restrictions (Supplementary Information, Supplementary Fig. 7). We used conditional auto-regressive (CAR) models[32] to assess the effects of the above factors on within- and cross-ZIP code transmission (Fig. 4). Specifically, for both within- and cross-ZIP code transmission, we fitted Poisson generalized linear mixed models (GLMM) with random effects and CAR priors to account for the inherent spatial-temporal autocorrelation in disease transmission data[32,33] (Supplementary Information, Supplementary Figs. 8–9).

We found that higher vaccination coverage and fewer POI visitors were associated with reduced non-household within- and cross-ZIP code transmission in the same week (Fig. 4). Estimates of coefficients are provided in Supplementary Table 2. The model identifies a strong effect of vaccination on SARS-CoV-2 transmission: during the early phase of vaccine rollout that aligns with the study period, a 12.5% newly vaccinated population was associated with reductions of 28.0% (95% CI: 14.0%–40.0%) and 14.8% (1.7%–26.4%) for within- and cross-ZIP code non-household transmission events, respectively. This marginal benefit may diminish for higher vaccine coverage as we expect the effect is nonlinear when the vaccinated population is near 100%. In contrast, a 78.1% increase of POI visitors per capita (ratio of the number of POI visitors to the population of each ZIP code) was associated with increases of 9.6% (0.3%–19.3%) and 14.4% (8.7%–20.2%) for within- and cross-ZIP code transmission outside households, respectively. In the foot traffic data, the POI category with the largest number of visitors

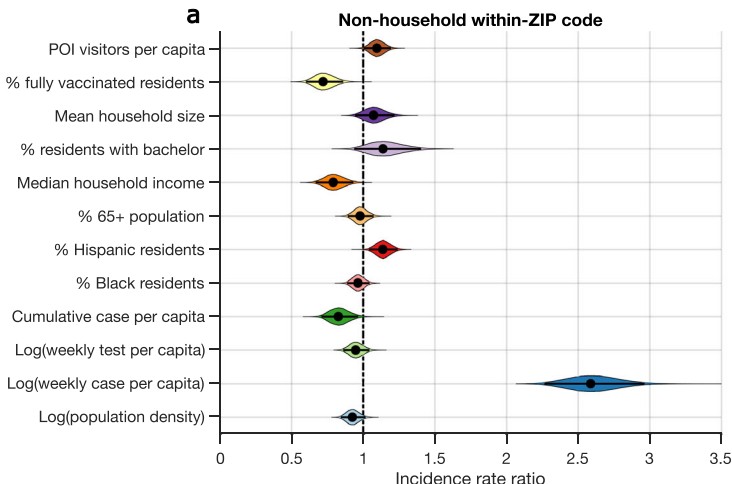
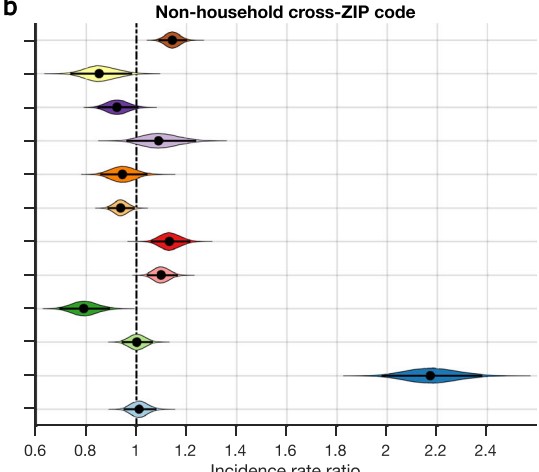

**Fig. 4 | Effects of various features on the transmission of SARS-CoV-2 in NYC.** Incidence rate ratios (exponentiated coefficients) for non-household within-ZIP code transmission and cross-ZIP code transmission are shown for 12 covariates in **a** and **b**, respectively (Deviance information criterion, DIC = 6342 for **a** and DIC = 12,644 for **b**). Coefficients were estimated using a Poisson generalized linear mixed model controlling for spatial-temporal autocorrelations. We used the log-transformed population as the offset in the regression model. Covariates were standardized and are shown on the y-axis. The incidence rate ratio quantifies the multiplicative change in the number of transmission events per each covariate increase of one standard deviation, controlling for other covariates. The violin plots show the distributions of incidence rate ratios. Black dots and horizontal black lines highlight the median estimates and 95% CIs. Distributions in **a** and **b** were obtained using $n = 20,000$ MCMC samples of the posterior estimates.

was restaurants and bars. It is possible, but not known, whether gathering in these places may contribute more to cross-ZIP code transmission than to within-ZIP code transmission. We further found that both within- and cross-ZIP code transmission had strong positive associations with log weekly cases per capita. A 13.5% increase of log weekly cases per capita was associated with increases of 158.8% (126.5%–196.4%) and 117.3% (97.7%–137.9%) for non-household within- and cross-ZIP code transmission. Higher percentage of Hispanic residents and lower cumulative cases per capita were associated with higher non-household transmission (see strength of effect in Supplementary Table 2). For cross-ZIP code transmission, cumulative cases per capita had a stronger effect than vaccination and POI visitors (Fig. 4b, Supplementary Table 2), indicating that prior infections may result in reduced cross-ZIP code transmission in locations with a higher attack rate. These findings reveal how health inequities related to COVID-19 manifest across NYC communities. Results also indicate that promoting vaccination and capacity limits or temporary limits on local businesses, schools, and other POIs in high-prevalence communities were effective in reducing SARS-CoV-2 transmission in NYC. These findings were corroborated with an alternate random-effect model (Supplementary Information) and testing of effect lags of one week and two weeks (Supplementary Figs. 10–12). Findings were also found robust to possible reduced response rate in contact tracing among children and elderly (Supplementary Fig. 13).

## Discussion

Here, leveraging detailed test and tracing data, we performed an analysis of ZIP code level SARS-CoV-2 transmission in NYC. The observed heterogeneity of SARS-CoV-2 spread at community scales implies that NPIs focusing on neighborhoods with extensive community transmission could potentially be more cost-effective. However, because communities with high test positivity were typically high poverty areas[3], during isolation and quarantine resources (such as food delivery, medication delivery, and access to safe isolation places) should be provided to address the disproportionate impact of the pandemic on these communities. Our statistical analyses suggest that the combination of vaccination and reactive, zone-based intervention measures implemented in NYC likely reduced the spread of COVID-19 during the second wave. There is evidence showing that COVID-19

vaccines can reduce transmission of SARS-CoV-2[34–37], although such effect has diminished with the emergence of more recent variants[38,39]. In the meantime, COVID-19 vaccine acceptance was found to be correlated with perception of risk and other psychological characteristics that may decrease the risk of transmission[40]. As a result, the overall effect of vaccination is possibly driven by the combined direct effect of transmission reduction and behavioral factors that correlate with vaccination coverage.

Our study found that the number of POI visitors is associated with both within- and cross-ZIP code transmission. As people travel for different reasons, it is critical to identify the types of travel that should be targeted by NPIs to reduce disease transmission. For instance, individuals working in essential businesses or emergency services may not be able to reduce movement, whereas individuals who travel long distances for resources might be better served by delivery or relocation of resources. In future outbreaks of respiratory infections, settings with increased infection risk should be first targeted through NPIs. Further studies are needed to identify the specific settings and behaviors for more precise interventions that simultaneously minimize disturbance to the society.

This study has several limitations. Firstly, the contact tracing data were biased to household exposure, and voluntarily reported close contacts, especially outside the household, were incomplete. As a result, identified clusters of exposure and transmission are largely confined to small networks, limiting the detection of complete transmission networks, including super-spreading events. Such bias is further compounded by differential reporting rate across age groups. However, the spatial transmission pattern is less affected by the selection bias if such bias is similar across ZIP code areas. Secondly, some communities may have a lower response rate to the calls from tracers. Further studies are needed to quantify the factors associated with the lower response rate for improving future contact tracing effectiveness. Thirdly, due to missing and incorrect personal identifying information, the matching to close contacts and their test results may be incomplete. Lastly, foot traffic data may have bias among POI categories and different age-groups. For instance, school-age children under 13 years old and other individuals without access to smart phones are not represented in the data.

With the global circulation of new variants of concern, such as Omicron and its sublinages[41], our findings can inform control management in other urban settings beyond NYC. Specifically, public health authorities should consider the community-level spatial dispersion of SARS-CoV-2 when designing control tactics, which can be analyzed in real time using contact tracing data. During the early stage of an emerging outbreak, contact tracing data may not be sufficient to support real-time analysis. However, once routine contact tracing is set up, it can support subsequent spatial analyses in real time if there is prevalent community transmission. Our analysis on the exposure network may inform a better definition of the proper geographical units for observation and interventions based on actual human interactions and disease transmission in NYC and elsewhere. Coordinated interventions targeting identified clusters of ZIP codes currently supporting the spatial transmission of SARS-CoV-2 could potentially produce more effective outbreak control. The findings may also support future pandemic preparedness and response. Further, the spatial transmission patterns might inform control policy for other respiratory pathogens sharing similar transmission routes. The operational performance of contact tracing can be used as a benchmark in urban settings and support modeling studies[42–45] of the potential effects of contact tracing on emerging infectious disease containment.

## Methods

### Data

We used contact tracing data collected in NYC from 1 October 2020 to 10 May 2021. The study period spans the second pandemic wave of COVID-19 in NYC. The data contain 5,735,726 phone call records of interactions between contact tracers and confirmed/probable cases and their contacts, as well as information gathered during the phone calls. Age and zip code of home location are available for most cases and contacts. Index cases and their contacts were identified in the dataset using a matching algorithm based on personal identifying information (see Supplementary Information). Use of this dataset in this study was approved by Columbia University Institutional Review Board (IRB) AAAT2182. Informed consent was obtained during the phone calls between contact tracers and participants prior to the collection of contact tracing information.

Demographic and socioeconomic data for NYC zip code tabulation areas (ZCTA) were compiled from the 5-year American Community Survey (ACS) (https://www.census.gov/programs-surveys/acs/data.html). Variables include population size, population density (persons per square kilometer), percentage of Black residents, percentage of Hispanic residents, percentage of population over 65 years old, median household income, percentage of residents with bachelor's degree, and mean household size. We downloaded the 2019 estimates for these variables using the R package tidycensus[46].

COVID-19 surveillance data in NYC at the MOZCTA (modified ZIP code tabulation area) level are available at the GitHub repository maintained by the NYC Department of Health and Mental Hygiene (DOHMH) (https://github.com/nychealth/coronavirus-data). We used weekly cases per capita, weekly tests per capita, and percentage of tests positive. Vaccination data were obtained from the public repository of DOHMH (https://github.com/nychealth/covid-vaccine-data). Human mobility data recording the weekly number of visitors to points of interest (POIs) in NYC were provided by SafeGraph (https://safegraph.com/), which aggregates anonymized location data from numerous mobile phone applications to provide insights about physical places, via the SafeGraph Community. To enhance privacy, Safe-Graph excludes census block group information if fewer than five devices visited an establishment in a month from a given census block group. We aggregated the mobility data to zip code level to estimate the weekly number of visitors (regardless of visitors' location of residence) to POIs in each zip code area. In the statistical analysis, we mapped the ACS data from the ZCTA level to the MOZCTA level to

align the scale of the data. The mapping between ZCTA and MOZCTA is available at https://data.cityofnewyork.us/Health/Modified-Zip-Code-Tabulation-Areas-MODZCTA-/pri4-ifjk.

### Reconstructing transmission networks

Due to asymptomatic and pre-symptomatic shedding, the reporting dates of index cases and contacts cannot be used to determine the direction of transmission. To address this issue, we developed a maximum-likelihood method to reconstruct transmission chains based on the risk of COVID-19 spread across different age groups. This approach includes three steps: (1) Estimate the infection time using symptom onset date or specimen collection date. Use the estimated infection time to determine the direction of exposure and transmission. (2) Estimate the probability of transmission for exposures across age groups using test and trace data. (3) Sample an ensemble of possible transmission networks and select the one that maximizes the transmission likelihood. Data analysis was performed using MATLAB R2021a. Details are provided in Supplementary Information.

### Statistical analysis

We used conditional autoregressive (CAR) models to analyze non-household within- and cross-ZIP code transmission in two separate models. The CAR model was implemented in a Bayesian hierarchical framework. Specifically, we fitted a Poisson generalized linear mixed model (GLMM) where the random effect was modeled by CAR priors to account for the inherent spatial-temporal autocorrelation present in the disease transmission data.

We modeled the numbers of non-household within- and cross-ZIP code transmission events using a modified Poisson generalized linear mixed model. Denote $y_{within}(i,t)$ and $y_{cross}(i,t)$ as the weekly numbers of non-household within-ZIP code and cross-ZIP code transmission events in ZIP code $i$ and week $t$. Here cross-ZIP code transmission events include both directions, i.e., transmission for which either infector or infectee lived in a certain ZIP code. The week for transmission is determined by the self-reported contact time between index cases and contacts. Fixed effects include log-transformed population density, log-transformed weekly cases per capita, log-transformed weekly tests per capita, cumulative cases per capita, percentage of Black residents, percentage of Hispanic residents, percentage of population over 65 years old, median household income, percentage of residents with a bachelor's degree, mean household size, percentage of fully vaccinated residents, and number of POI visitors per capita. All covariates were standardized to have mean zero and standard deviation one. We used log-transformed population as an offset, assuming the numbers of both within-ZIP code and cross-ZIP code transmission events are proportional to local population. In the regression model, we used the weekly case per capita to represent the local force of infection that impacts the number of observed within-ZIP code transmission events.

Specifically, the model for non-household within-ZIP code transmission is described by the following equation:

$$
\begin{aligned}
\log(y_{within}&(i,t+d)) \\
= \log&(population(i)) + \beta_1 \times \log(population\,density(i)) + \beta_2 \\
&\times \log(weekly\,cases\,per\,capita(i,t)) + \beta_3 \\
&\times \log(weekly\,tests\,per\,capita(i,t)) + \beta_4 \\
&\times cumulative\,cases\,per\,capita(i,t) + \beta_5 \times \%\,Black\,resident(i) + \beta_6 \\
&\times \%\,Hispanic\,resident(i) + \beta_7 \times \%\,resident\,over\,65(i) + \beta_8 \\
&\times median\,household\,income(i) + \beta_9 \times \%\,bachelor's\,degree(i) + \beta_{10} \\
&\times mean\,household\,size(i) + \beta_{11} \times \%\,fully\,vaccinated\,resident(i,t) + \beta_{12} \\
&\times weekly\,POI\,visitors\,per\,capita(i,t) + \psi_{it} + \varepsilon_{it}.
\end{aligned}
\tag{1}
$$

Here $d$ is the lag (in weeks), $\log(population(i))$ is the offset, $\psi_{it}$ is the random effect for location $i$ and week $t$, and $\varepsilon_{it}$ is the error term. In the main model, we used $d = 0$ (no lag). We additionally tested $d = 1$ and $d = 2$ as a sensitivity analysis.

The model for cross-zip code transmission is defined similarly:

$$
\begin{aligned}
&\log(y_{cross}(i, t+d)) \\
&= \log(population(i)) + \beta_1 \times \log(population\ density(i)) + \beta_2 \\
&\quad \times \log(weekly\ cases\ per\ capita(i,t)) + \beta_3 \\
&\quad \times \log(weekly\ tests\ per\ capita(i,t)) + \beta_4 \\
&\quad \times cumulative\ cases\ per\ capita(i,t) + \beta_5 \times \%\ Black\ resident(i) + \beta_6 \\
&\quad \times \%\ Hispanic\ resident(i) + \beta_7 \times \%\ resident\ over\ 65(i) + \beta_8 \\
&\quad \times median\ household\ income(i) + \beta_9 \times \%\ bachelor's\ degree(i) + \beta_{10} \\
&\quad \times mean\ household\ size(i) + \beta_{11} \times \%\ fully\ vaccinated\ resident(i,t) + \beta_{12} \\
&\quad \times weekly\ POI\ visitors\ per\ capita(i,t) + \psi_{it} + \varepsilon_{it}.
\end{aligned} \tag{2}
$$

We implemented the model using the function ST.CARar in the R package CARBayesST. Using a Bayesian hierarchical framework, model coefficients and parameters were estimated using a Markov chain Monte Carlo (MCMC) algorithm. We fitted the model using data from 177 MOZCTAs and 31 weeks. Statistical analysis was performed using R statistical software version 4.1.0. Details on model implementation, evaluation of spatial-temporal autocorrelation in residues, and sensitivity analysis are provided in Supplementary Information.

### Reporting summary
Further information on research design is available in the Nature Research Reporting Summary linked to this article.

## Data availability
COVID-19 surveillance data in NYC at the MOZCTA (modified ZIP code tabulation area) level are publicly available at the GitHub repository maintained by the NYC Department of Health and Mental Hygiene (DOHMH) (https://github.com/nychealth/coronavirus-data). Demographic and socioeconomic data for NYC zip code tabulation areas (ZCTA) are available from the 5-year American Community Survey (ACS) (https://www.census.gov/programs-surveys/acs/data.html). Contact tracing records and individual testing results are subject to restrictions for the protection of patient privacy. Requests for data access should be addressed to NYC DOHMH and NYC Health + Hospitals or the corresponding author. The corresponding author will respond to requests within two weeks and facilitate communications with NYC DOHMH and NYC Health + Hospitals, who will provide details of any restrictions imposed on data use via data use agreements.

## Code availability
Custom code and data supporting the statistical analysis are publicly available at GitHub (https://github.com/SenPei-CU/NYC_contacttracing)[47].

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

## Acknowledgements

This study was supported by funding from the National Institutes of Health grant R01AI163023, Centers for Disease Control and Prevention U01CK000592 and 75D30122C14289, National Science Foundation DMS-2229605, Council of State and Territorial Epidemiologists NU38OT00297 and a gift from the Morris-Singer Foundation. We thank Sharon Greene, Celia Quinn, Hannah Helmy and Jeffrey Sachs for comments and discussions. We thank Shigeru Odani in the T2 data and analytics team for assistance in data analysis. We also thank SafeGraph for providing foot traffic data.

## Author contributions

S.P., T.L., and J.S. conceived the study and managed the project, S.P., S.K., J.C., and S.F. performed the analysis, S.F., C.T., J.B., S.D.A., K.B., J.K.V., and T.L. curated data, S.K., J.C., W.Y., S.F., C.T., J.B., S.D.A., K.B., J.K.V., T.L., and J.S. investigated the results, S.P. drafted the manuscript, all authors revised and reviewed the manuscript.

## Competing interests

J.S. and Columbia University disclose partial ownership of SK Analytics. J.S. discloses consulting for BNI. All other authors declare no competing interests.

## Additional information

**Correspondence and requests** for materials should be addressed to Sen Pei.

