## [Peer Review File · Nature Communications]

Contact tracing reveals community transmission of COVID-19 in New York CityREVIEWER COMMENTS

Reviewer #1 (Remarks to the Author):

In the manuscript titled "Contact tracing reveals community transmission of COVID-19 in New York City" the authors analyzed the data from an expansive contact tracing effort carried out by the NYC Test & Trace Corps during the second epidemic wave in the New York City. Specifically, the authors described the topological features/spatial dispersion pattern of the contact exposure as well as the transmission networks revealed by the contact tracing. The authors also performed regression analysis to identify drivers of both within-ZIP code and across-ZIP code SARS-CoV-2 transmission. This is a nicely conducted study over a very unique dataset. I have a few comments below:

- 1) Could the authors describe the variants that were circulating during the period of study?
- 2) Could the authors calculate the secondary attack rate among tested contacts? Could the authors differentiate the types of close contacts (i.e. household, work, school, leisure etc.)? It would be interesting to investigate if contact types influence the risk of transmission.
- 3) The lead authors previously used mathematical models to investigate the under-reporting & the cryptic spread of SARS-CoV-2 during the first epidemic wave in NYC. Can the authors evaluate the extent of under-reporting during this epidemic wave? Will the findings be biased due to under-reporting?
- 4) For cross-ZIP code transmission events, it wasn't clear if the authors were evaluating the transmission originated from a ZIP code level location (i.e. the infectors were in ZIP code i) or the other way around (i.e. the infectee were in ZIP code i). Please clarify.
- 5) For regression in Figure 4, why introducing $\log(\text{population})$ as a fixed offset?
- 6) For regression in Figure 4, isn't that the dependent variable of "within-ZIP code" transmission" a subset of "weekly case per capita"? I don't quite understand why this "weekly case per capita" need to be included as an independent variable. I would appreciate if the authors could explain their motivations of including this term.
- 7) In Figure 4, the authors found that higher vaccination rate was associated with both within-ZIP code and cross-ZIP code SARS-CoV-2 transmission. Can the authors present the distribution of vaccination rates at ZIP Code level? Public data suggest that the vaccination rates at state-level yet to reach high level ($\sim 20\%$ fully vaccinated) during the study period and it would be interesting to understand the spatial heterogeneity of vaccination at finer scale. Would the authors think the "protective" effect from high vaccination rate a direct effect caused by vaccination or vaccination at the early stage of the vaccine roll-out was correlated with prudent/protective behavior that were actually a causal mediator of the risk reduction?

Reviewer #2 (Remarks to the Author):

This study explores the SARS-CoV-2 transmission clusters at the community level and assesses the relationship between potential factors, such as vaccination coverage and human mobility, and the virus transmission through conducting spatial analysis using the contact tracing data. The results indicate the feasibility of establishing strategies at the community level for controlling the spread of SARS-CoV-2 in New York. It also provided guidance for taking possible measures to inhibit community transmission in NY. Overall, this paper is well-structured and detailed in explaining the findings and methods. Though there are a few suggestions as follows. I only have one major concern about the validity of imputation of infection time, since this may change other results based on the estimated transmission network.

Major comments:

1. The author using the incubation period distribution to sample the infection time from onset time, which would only be correct when the outbreak is stable (with growth rate = 0), see (<https://www.pnas.org/doi/10.1073/pnas.2011548118>). More work is needed to be done, to show that the current approach about imputation of infection time is correct.
2. As acknowledged that the quality of the contact tracing data would be important, and therefore is it possible to show that most of the results would be robust to this? E.g. what if children/elderly is less likely to report and how may this impact the conclusion?

Minor comments:

1. The interview data used in this retrospective study were collected during the second pandemic wave in NYC. However, in practice, the contact tracing data should be collected in succession. If using contact tracing data to do real-time analysis for future pandemic preparedness as the authors suggest, is it possible that the data are insufficient for giving reliable results of spatial analysis during the early outbreak?

2. Abstract: 'We find that higher vaccination coverage and reduced numbers of visitors to points-of-interest are associated with fewer within- and cross-ZIP code transmission events': what is the direction of association? Higher or lower?

3. Results: for subsection 'Contact tracing in NYC', there should be a paragraph about the contact tracing system. E.g. if it is electronic, since this may have impact of reporting like age. For subsection 'Exposure and transmission networks', there should be a paragraph about the fitted transmission network construction.

4. Results: 'a 12.48% newly vaccinated population was associated with reductions of 28.0% (95% CI: 14.0% – 40.0%) and 14.8% (1.7% – 26.4%) for within- and cross-ZIP code non-household transmission events, respectively.' It seems difficult to interpret from me, since maximum of vaccine coverage is 100%, and we expect non-linear effect (e.g. decreasing marginal return or indirect protection).

5. Supplementary information: The authors can briefly introduce the Moran's I test, like adding a description of how to develop the Moran's I test and what the hypotheses are. It would be easier for readers who are without this background knowledge to understand the results in Extended data Figure 4.

6. Figure 3 panel a and b: it is difficult to identify the direction of arrows, maybe change a color for the heads of arrow?

Reviewer #3 (Remarks to the Author):

Overview:

This is a very interesting study using granular contact tracing data in a dense urban setting. The methods are well described, and the prose is clear. I have a few minor comments and questions.

Comments:

Main Manuscript:

- Line 84: I think the reference should be to Fig 1.c instead of Fig 1.d.
- Line 104: Regarding cross-borough contacts. Human mobility data often shows "movement" or "connections" between large admin regions, however, at higher granularity these contacts end up being local (as described a bit in line 140). Are the cross-borough connections that you see primarily driven by contacts between adjacent neighborhoods at borough borders or by transitions between boroughs to points of interest? The answer may help better target your suggested interventions.
- Line 107: How were contacts lined to lab confirmed cases? The same matching algorithm described above?
- Line 151: Is "population mobility" here visits to POIs or are you using other measures?
- Line 163: Is the mobility value normalized to a baseline? Would certain POIs (perhaps pharmacies) have an expected increase in traffic regardless of NPIs? Would you expect differential change in POI visitation driven by SES of different regions? How could the foot traffic to POIs (especially cross-borough) be affected by individuals who aren't able to work from home and who may need to travel to these locations for their job?
- Line 178: Cross-zip code transmission is more strongly associated with increase in visits to POIs than within-zip code transmission. What type of movement could this be (work related, seeking resources etc.) and is it targetable by NPIs? Depending on the type of movement, zip code based restrictions may be very challenging to implement.

- Line 188: Is the POI data available at higher granularity? It would be very useful to differentiate schools from businesses to identify appropriate policies.

- Discussion: I would appreciate a more detailed discussion of potential interventions. Why were people traveling? What types of travel should have been targeted to reduce transmission? For example, individuals working in emergency services, or those required to show up in person may not ever be able to reduce movement while individuals who travel long distances for resources could be better served by delivery or relocation of resources. Even with new surges NPIs are generally being lifted across the United States. What can we glean from this work that would help us better target responses and behaviors that can lead transmission?

Supplement:

- How successful was the matching by PII? Was this based on some identifier or other metrics such as fuzzy-matching names / birthdays?

- The network reconstruction mechanism is fascinating and draws well from existing research!

REVIEWER COMMENTS

Reviewer #1 (Remarks to the Author):

In the manuscript titled “Contact tracing reveals community transmission of COVID-19 in New York City” the authors analyzed the data from an expansive contact tracing effort carried out by the NYC Test & Trace Corps during the second epidemic wave in the New York City. Specifically, the authors described the topological features/spatial dispersion pattern of the contact exposure as well as the transmission networks revealed by the contact tracing. The authors also performed regression analysis to identify drivers of both within-ZIP code and across-ZIP code SARS-CoV-2 transmission. This is a nicely conducted study over a very unique dataset. I have a few comments below:

Response: We appreciate the reviewer’s positive feedbacks on this study.

1) Could the authors describe the variants that were circulating during the period of study?

Response: During the study period, the circulating strains of SARS-CoV-2 in NYC were dominated by the index virus strain (per WHO definition, genomic sequence of SARS-CoV-2 identified from the first cases in December 2019), but the Iota (B.1.526) and Alpha (B.1.1.7) variants gradually replaced the wild type during the spring of 2021. We report the variants circulating during the study period in the main text and have added a supplementary figure (Supplementary Fig. 2) in the SI showing the percentage of SARS-CoV-2 variants in NYC from Jan 2021 to May 2021. Please find change in lines 84-87 of the revised manuscript.

2) Could the authors calculate the secondary attack rate among tested contacts? Could the authors differentiate the types of close contacts (i.e. household, work, school, leisure etc.)? It would be interesting to investigate if contact types influence the risk of transmission.

Response: Thanks for the suggestion. The positivity rate among all tested exposures is 55.8%. However, due to selection bias (infected individuals were more likely to seek tests), the actual secondary attack rate should be lower. We further disaggregated testing results for exposures of different types (healthcare facility contact, home health aide, household member, intimate partner, large gathering contact, other close proximity, workplace contact) (see Supplementary Fig. 3). The positivity rate is highest for household member (58.7%) and lowest for workplace contact (20.3%). If selection bias is the same across exposure types, household contact should have the highest secondary attack rate. We now include this result in the revised manuscript (lines 91-96).

3) The lead authors previously used mathematical models to investigate the under-reporting & the cryptic spread of SARS-CoV-2 during the first epidemic wave in NYC. Can the authors evaluate the extent of under-reporting during this epidemic wave? Will the findings be biased due to under-reporting?

Response: Evaluating under-reporting over space and time is a challenging task, particularly at fine geographical scales such as the ZIP code level. We did not attempt to estimate under-reporting in this study as the focus here is the contact tracing data. However, to examine whether the findings are robust to under-reporting, we have performed additional sensitivity analyses. Specifically, we randomly removed 50% of the contact tracing records reported by individuals <18 years old or >65 years old, representing the scenario that children and elderly are less likely to report their close contacts. The distributions of exposure and transmission clusters in Fig. 2 remain similar, so does the spatial transmission of SARS-CoV-2 in NYC in Figure 3. Further, results of the regression model remain similar to the main model (see Supplementary Fig. 13). We also list under-reporting as a limitation of the data in the discussion section. Please find changes in lines 229-231 and 265-266 of the revised manuscript.

4)For cross-ZIP code transmission events, it wasn't clear if the authors were evaluating the transmission originated from a ZIP code level location (i.e. the infectors were in ZIP code i) or the other way around (i.e. the infectee were in ZIP code i). Please clarify.

Response: We thank the reviewer for pointing this out. Cross-ZIP code transmission events include both directions, i.e., transmission for which either infector or infectee lived in a certain ZIP code. This is now clarified in lines 189-190 of the revised manuscript.

5)For regression in Figure 4, why introducing $\log(\text{population})$ as a fixed offset?

Response: We used log-transformed population as an offset, assuming the numbers of both within-ZIP code and cross-ZIP code transmission events are proportional to local population. We clarify this assumption in the subsection 3.1 (statistical model) in the SI.

6)For regression in Figure 4, isn't that the dependent variable of "within-ZIP code" transmission" a subset of "weekly case per capita"? I don't quite understand why this "weekly case per capita" need to be included as an independent variable. I would appreciate if the authors could explain their motivations of including this term.

Response: This is a good question. In the regression model, we used the weekly cases per capita to represent the local force of infection that impacts the number of observed within-ZIP code transmission events. While within-ZIP code transmission events are a subset of weekly cases, the observed transmission events are much fewer and may deviate from the pattern of weekly case per capita due to reporting bias and other factors. We therefore decided to include weekly case per capita as a covariate in the regression model. This is now explained in the subsection 3.1 (statistical model) in the SI.

7)In Figure 4, the authors found that higher vaccination rate was associated with both with-ZIP code and cross-ZIP code SARS-CoV-2 transmission. Can the authors present the distribution of vaccination rates at ZIP Code level? Public data suggest that the vaccination rates at state-level yet to reach high level (~20% fully vaccinated) during the study period and it would be interesting to understand the spatial heterogeneity of vaccination at finer scale. Would the authors think the "protective" effect from high vaccination rate a direct effect caused by vaccination or vaccination at the early stage of the vaccine roll-out was correlated with prudent/protective behavior that were actually a causal mediator of the risk reduction?

Response: Thank you for the question and suggestion. For vaccination coverage, we observed large spatial heterogeneity at the ZIP code level (Supplementary Fig. 6). The percentage of the population fully vaccinated ranged from 22.7% to 82.8% during the week of May 22, 2021. In the revised SI, we add a supplementary figure to visualize the spatial heterogeneity of vaccination at the ZIP code scale.

There is evidence showing that COVID-19 vaccines can reduce transmission of SARS-CoV-2¹⁻⁴, although this effect appears to have diminished following the emergence of new variants^{5,6}. At the same time, COVID-19 vaccine acceptance was found correlated with perception of risk and other psychological characteristics that may decrease the risk of transmission⁷. As a result, the overall effect of vaccination is possibly driven by the combined direct effect of transmission reduction and behavioral factors that correlate with vaccination coverage. We have added discussion of this point to the revised manuscript. Please find changes in lines 244-249.

Reviewer #2 (Remarks to the Author):

This study explores the SARS-CoV-2 transmission clusters at the community level and assesses the relationship between potential factors, such as vaccination coverage and human mobility, and the virus transmission through conducting spatial analysis using the contact tracing data. The results indicate the feasibility of establishing strategies at the community level for controlling the spread of SARS-CoV-2 in New York. It also provided guidance for taking possible measures to inhibit community transmission in NY. Overall, this paper is well-structured and detailed in explaining the findings and methods. Though there are a few suggestions as follows. I only have one major concern about the validity of imputation of infection time, since this may change other results based on the estimated transmission network.

Response: We appreciate the reviewer's comments and constructive suggestions to improve the manuscript.

Major comments:

1. The author using the incubation period distribution to sample the infection time from onset time, which would only be correct when the outbreak is stable (with growth rate = 0), see (<https://www.pnas.org/doi/10.1073/pnas.2011548118>). More work is needed to be done, to show that the current approach about imputation of infection time is correct.

Response: We thank the reviewer for pointing out this relevant study. Indeed, incubation period distribution estimated using contact tracing data may be subject to bias depending on outbreak dynamics and the method used to calculate the delay distribution. As indicated in Park et al., typically, within-individual delay distributions such as the incubation period are not affected by epidemic dynamics if a forward delay distribution is used (i.e., a distribution of incubation period from a cohort of infected individuals that were infected at the same time). However, the incubation period could be biased if the backward delay distribution is used (i.e., a distribution of incubation period from a cohort of infected individuals that developed symptoms at the same time). Specifically, the backward incubation period is biased low (high) when the epidemic is growing (declining). To reduce the effect of such potential bias on the network reconstruction, we used the incubation period distribution estimated in Hu et al. In this study, the authors argued that the contact tracing data were collected from in-depth epidemiological investigations, allowing robust estimation of key time-to-event distributions. Moreover, the exponential growth phase of the outbreak lasted only about two weeks (thus potential underestimation of the incubation period is limited) and the effort heavily relied on forward contact tracing. The effect of the potential bias of incubation period on network reconstruction is therefore limited.

To verify the robustness of inference, we performed sensitivity analyses using two alternative incubation period distributions – one with a 10% underestimation of the mean ($k = 1.58$, $\lambda = 6.40$, mean 5.76 days) and another one with a 10% overestimation of the mean ($k = 1.58$, $\lambda = 7.82$, mean 7.04 days). We used the same shape parameter k and varied the scale parameter λ to adjust the mean incubation period. For each pair of index case and reported exposure with symptom onsets, we drew 1,000 samples of infection times using the incubation period distribution. Denote p_{\rightarrow} as the fraction of samples for which the inferred transmission direction is from the index case to exposure. We computed $\{p_{\rightarrow}\}$ for all pairs and calculated the change of $\{p_{\rightarrow}\}$ due to biased incubation period distributions. For the underestimated incubation period, the median change is -0.003 (95% CI: [-0.049, 0.046]); for the overestimated incubation period, the median change is 0.003 ([-0.044, 0.047]). This experiment indicates that potential bias in the incubation period distribution does not dramatically affect inference of transmission direction. We have added a detailed discussion on this point in the revised SI and mention this additional sensitivity analysis in the main text (lines 139-141).

2. As acknowledged that the quality of the contact tracing data would be important, and therefore is it

possible to show that most of the results would be robust to this? E.g. what if children/elderly is less likely to report and how may this impact the conclusion?

Response: Thanks for the constructive suggestion. In the revised manuscript, we perform an additional sensitivity analysis to test whether the findings in the regression model are robust to potential different response rates in contact tracing among age groups. Specifically, we randomly removed 50% of the contact tracing records reported by individuals <18 years old or >65 years old, representing the scenario that children and elderly are less likely to report their close contacts. The distributions of exposure and transmission clusters in Fig. 2 remain similar, so does the spatial transmission of SARS-CoV-2 in NYC in Figure 3. Further, results remain similar to the main model (Supplementary Fig. 13). We also mention this potential bias as a limitation in the discussion section. Please find changes in the revised SI and lines 229-231, 265-266 in the main text.

Minor comments:

1. The interview data used in this retrospective study were collected during the second pandemic wave in NYC. However, in practice, the contact tracing data should be collected in succession. If using contact tracing data to do real-time analysis for future pandemic preparedness as the authors suggest, is it possible that the data are insufficient for giving reliable results of spatial analysis during the early outbreak?

Response: Indeed, the contact tracing data collected during the early phase of an outbreak may not be sufficient to support robust spatial analysis. Once routine contact tracing is set up, it can support subsequent spatial analysis in real time if there is prevalent community transmission. The example in NYC during the COVID-19 pandemic shows that such analysis is feasible. In addition, the spatial transmission pattern of SARS-CoV-2 can still inform control policy for other respiratory pathogens sharing similar transmission routes. In the revised manuscript, we have added a discussion of this issue in lines 279-281 and 287-288.

2. Abstract: 'We find that higher vaccination coverage and reduced numbers of visitors to points-of-interest are associated with fewer within- and cross-ZIP code transmission events': what is the direction of association? Higher or lower?

Response: We rephrase this sentence to "We find that locations with higher vaccination coverage and lower numbers of visitors to points-of-interest have reduced within- and cross-ZIP code transmission events". We hope this clarifies the direction of association.

3. Results: for subsection 'Contact tracing in NYC', there should be a paragraph about the contact tracing system. E.g. if it is electronic, since this may have impact of reporting like age. For subsection 'Exposure and transmission networks', there should be a paragraph about the fitted transmission network construction.

Response: Thank you for the suggestion. Contact tracing was performed through phone calls and text messages, which can reach most residents of NYC. We now include a more detailed introduction of the contact tracing workflow in the revised manuscript (lines 72-79). We have also added a description of the transmission network construction process in the main text. Please find these changes in lines 130-141.

4. Results: 'a 12.48% newly vaccinated population was associated with reductions of 28.0% (95% CI: 14.0% – 40.0%) and 14.8% (1.7% – 26.4%) for within- and cross-ZIP code non-household transmission events, respectively.' It seems difficult to interpret from me, since maximum of vaccine coverage is 100%, and we expect non-linear effect (e.g. decreasing marginal return or indirect protection).

Response: This is a good point. The result was obtained from a linear regression model for the period when vaccine coverage was not close to 100%. In the revised manuscript, we highlight that this result is only valid during the early phase of vaccine rollout. We also stress that the marginal benefit may diminish for higher vaccine coverage as we expect the effect is nonlinear when the vaccinated population is close to 100%. Please find changes in lines 205-210.

5. Supplementary information: The authors can briefly introduce the Moran's I test, like adding a description of how to develop the Moran's I test and what the hypotheses are. It would be easier for readers who are without this background knowledge to understand the results in Extended data Figure 4.

Response: Good suggestion. We now have added introduction of the Moran's I test and the Durbin-Watson test in the SI. Please find details in the revised SI, subsection 3.2.

6. Figure 3 panel a and b: it is difficult to identify the direction of arrows, maybe change a color for the heads of arrow?

Response: Thank you for the suggestion. We tried changing the color of the arrow heads, but the effect is not satisfactory. We therefore removed exposure links with less than 30 events and transmission links with less than 2 events from the maps in order to better visualize the directions of major transmission links. We hope the direction of the arrows is now clearer.

Reviewer #3 (Remarks to the Author):

Overview:

This is a very interesting study using granular contact tracing data in a dense urban setting. The methods are well described, and the prose is clear. I have a few minor comments and questions.

Response: We appreciate the reviewer's positive feedback.

Comments:

Main Manuscript:

- Line 84: I think the reference should be to Fig 1.c instead of Fig 1.d.

Response: Thanks for finding the typo. This is now fixed in the revised manuscript.

- Line 104: Regarding cross-borough contacts. Human mobility data often shows "movement" or "connections" between large admin regions, however, at higher granularity these contacts end up being local (as described a bit in line 140). Are the cross-borough connections that you see primarily driven by contacts between adjacent neighborhoods at borough borders or by transitions between boroughs to points of interest? The answer may help better target your suggested interventions.

Response: Good question. We performed an additional analysis and find that the majority of cross-ZIP code transmission events were not between neighboring ZIP code areas. Among these cross-ZIP code transmission events, only 2,536 (32.4%) occurred between neighboring ZIP code areas, indicating that the majority of cross-ZIP code transmission contributed to non-local disease spread. Among 2,187 cross-borough transmission events, only 48 (2.2%) were between neighboring ZIP code areas. We present these findings in the revised manuscript in lines 156-160.

- Line 107: How were contacts lined to lab confirmed cases? The same matching algorithm described above?

Response: Exactly. The DOHMH and T2 team used a matching algorithm validated by a hand-labeled gold-standard data set. A detailed description of the matching algorithm has been added in the revised SI. We also rephrase the sentence to clarify the process. Please see changes in lines 123-124 of the revised manuscript.

- Line 151: Is “population mobility” here visits to POIs or are you using other measures?

Response: Yes. In the revised manuscript, we have changed “population mobility” to “the number of individuals visiting points-of-interest”. Please see lines 178-180.

- Line 163: Is the mobility value normalized to a baseline? Would certain POIs (perhaps pharmacies) have an expected increase in traffic regardless of NPIs? Would you expect differential change in POI visitation driven by SES of different regions? How could the foot traffic to POIs (especially cross-borough) be affected by individuals who aren't able to work from home and who may need to travel to these locations for their job?

Response: Those are good questions. In the regression model, we used the raw number of POI visitors in each ZIP code area (not normalized). Over the course of the pandemic, the change in POI visitation varied across different categories. For instance, the recovery for educational service was faster than for restaurant & bar and grocery & pharmacy (Supplementary Fig. 7). Previous work has linked the differential change in POI visitation with socioeconomic status, finding that disadvantaged groups were not able to reduce their mobility as sharply, and that the POIs that they visit are more crowded and are therefore associated with higher risk⁸. As the data on individuals' jobs are less reliable, it is challenging to evaluate how occupancy can affect POI visitation in this study. In the revised manuscript, we add a discussion in the revised SI, subsection 3.1.

- Line 178: Cross-zip code transmission is more strongly associated with increase in visits to POIs than within-zip code transmission. What type of movement could this be (work related, seeking resources etc.) and is it targetable by NPIs? Depending on the type of movement, zip code based restrictions may be very challenging to implement.

Response: In the foot traffic data, the POI category with the largest number of visitors is restaurants and bars. It is possible, but not known, whether gathering in these places may contribute more to cross-ZIP code transmission than to within-ZIP code transmission. These settings are targetable with NPIs. In fact, restaurants and bars were among major business targeted with the interventions in NYC. We agree that restrictions based on POI categories is more precise than those based on ZIP code. More studies are needed to identify POI categories that should be targeted. In the revised manuscript, we have added a discussion of these issues in lines 213-215 and 251-259.

- Line 188: Is the POI data available at higher granularity? It would be very useful to differentiate schools from businesses to identify appropriate policies.

Response: The foot traffic data include information about the types of POIs, represented by the NAICS (North American Industry Classification System) code. However, certain POI categories may be over-represented in the dataset as location information was collected using voluntary check-in records in mobile phone apps. In addition, school-age children under 13 years old and other individuals without access to smart phones are not included in the data. In the revised manuscript, we discuss these data limitations (lines 271-273, SI subsection 3.1) and suggest further research.

- Discussion: I would appreciate a more detailed discussion of potential interventions. Why were people traveling? What types of travel should have been targeted to reduce transmission? For example, individuals working in emergency services, or those required to show up in person may not ever be able to reduce movement while individuals who travel long distances for resources could be

better served by delivery or relocation of resources. Even with new surges NPIs are generally being lifted across the United States. What can we glean from this work that would help us better target responses and behaviors that can lead transmission?

Response: Thank you for this suggestion. These questions are important, relevant to disease control practice, and worthy of further discussion. Capacity limit on certain business (e.g., restaurants, bars, gyms, entertainment venues, etc.) in communities with high test positivity rates may be effective in reducing transmission in NYC, possibly due to decreasing the elevated infection risk in those settings. In future outbreaks of respiratory infections, settings with increased infection risk should be first targeted through NPIs; however, further studies are needed to identify specific settings and behaviors allowing more precise interventions that simultaneously minimize disturbance to the society. We have added a paragraph on these issues to the discussion section, including mention of future research questions that might be prioritized. Please find these changes in lines 251-259 of the revised manuscript.

Supplement:

- How successful was the matching by PII? Was this based on some identifier or other metrics such as fuzzy-matching names / birthdays?

Response: A detailed description of the matching algorithm is provided in the revised SI, section 1. The matching algorithm was validated using a gold-standard dataset created by subjectively hand-labeling a random sample of the total record. The sensitivity of the matching algorithm was 0.957, specificity was 0.702 and precision was 0.922. We now provide this information in the SI.

- The network reconstruction mechanism is fascinating and draws well from existing research!

Response: Thanks. We are glad to see the reviewer likes the approach.

Reference

1. Prunas, O. *et al.* Vaccination with BNT162b2 reduces transmission of SARS-CoV-2 to household contacts in Israel. *Science* **375**, 1151–1154 (2022).
2. Harris, R. J. *et al.* Effect of Vaccination on Household Transmission of SARS-CoV-2 in England. *New England Journal of Medicine* **385**, 759–760 (2021).
3. Shah, A. S. V. *et al.* Effect of Vaccination on Transmission of SARS-CoV-2. *New England Journal of Medicine* **385**, 1718–1720 (2021).
4. Stokel-Walker, C. What do we know about covid vaccines and preventing transmission? *BMJ* **376**, o298 (2022).
5. Singanayagam, A. *et al.* Community transmission and viral load kinetics of the SARS-CoV-2 delta (B.1.617.2) variant in vaccinated and unvaccinated individuals in the UK: a prospective, longitudinal, cohort study. *The Lancet Infectious Diseases* **22**, 183–195 (2022).

6. Eyre, D. W. *et al.* Effect of Covid-19 Vaccination on Transmission of Alpha and Delta Variants. *New England Journal of Medicine* **386**, 744–756 (2022).
7. Murphy, J. *et al.* Psychological characteristics associated with COVID-19 vaccine hesitancy and resistance in Ireland and the United Kingdom. *Nat Commun* **12**, 29 (2021).
8. Chang, S. *et al.* Mobility network models of COVID-19 explain inequities and inform reopening. *Nature* **589**, 82–87 (2021).

REVIEWERS' COMMENTS

Reviewer #1 (Remarks to the Author):

The authors have addressed all my previous concerns and I do not have further comments.

Reviewer #2 (Remarks to the Author):

Thank you for the response. I think this looks good and I don't have any further comments.

Reviewer #3 (Remarks to the Author):

The authors have sufficiently responded to my questions from the previous round of review.